# Graph Masked Language Models

## Abstract

Language Models (LMs) and Graph Neural Networks (GNNs) have shown great promise in their respective areas, yet integrating structured graph data with rich textual information remains challenging. In this work, we propose Graph Masked Language Models (GMLM), a novel dual-branch architecture that combines the structural learning of GNNs with the contextual power of pretrained language models. Our approach introduces two key innovations: (i) a semantic masking strategy that utilizes graph topology to selectively mask nodes based on their structural importance, and (ii) a soft masking mechanism that interpolates between original node features and a learnable mask token, ensuring smoother information flow during training. Extensive experiments on multiple node classification and language understanding benchmarks demonstrate that GMLM not only achieves state-of-the-art performance but also exhibits enhanced robustness and stability. This work underscores the benefits of integrating structured and unstructured data representations for improved graph learning.

## 1 Introduction

Graph Neural Networks (GNNs) Scarselli et al. (2009), Zhou et al. (2021) have demonstrated remarkable success in tasks such as node classification, link prediction, and graph classification. However, most existing GNN architectures primarily focus on structural information and local neighborhood features, potentially overlooking rich semantic relationships between nodes. While recent advances in natural language processing, particularly pre-trained language models like BERT Devlin et al. (2019), have revolutionized our ability to capture contextual information, their integration with graph learning remains largely unexplored.

Graph Attention Networks (GATs) Veličković et al. (2018) have shown promise in selectively aggregating neighborhood information through learnable attention mechanisms. However, their effectiveness can be limited by the challenge of capturing long-range dependencies and semantic relationships in graphs. Meanwhile, masked language modeling, a key innovation in BERT, has proven highly effective for learning contextual representations in text data. This raises an intriguing question: Can we adapt masked modeling techniques to enhance graph representation learning?

Traditional node masking approaches in GNNs Mishra et al. (2021) often employ random masking strategies or complete removal of features, which can lead to information loss and suboptimal learning. Furthermore, these approaches typically ignore the structural importance of nodes in the graph topology, potentially masking critical nodes that are essential for understanding the global graph structure.

In this paper, we present a novel framework that bridges these gaps by combining the structural learning capabilities of GATs with the contextual understanding of BERT-style masked modeling. Our approach introduces two key innovations: (1) a semantic masking strategy that considers node importance in the graph topology, and (2) a soft masking mechanism that allows for partial information flow during training. This combination enables our model to learn both structural and semantic patterns effectively, leading to more robust node representations.

Our main contributions of this work can be summarized as follows:

1. We propose a novel dual-branch architecture that effectively combines Graph Attention Networks with BERT-style masked modeling for enhanced node representation learning.

2. We introduce a semantic masking strategy that selectively masks nodes based on their structural importance in the graph, ensuring that the model learns from topologically significant nodes. We also develop a soft masking mechanism that creates interpolated node representations, allowing for gradual information flow during training and better preservation of feature relationships Fig. 1.

3. We design a multi-layer fusion network that effectively combines structural and semantic information from the graph and language model branches.

## 2 Related Works

Graph Neural Networks (GNNs) Zhou et al. (2021) provide a powerful framework for learning from graph-structured data. Building upon early spectral and spatial convolutional methods, Graph Attention Networks (GATs) Veličković et al. (2018) integrated self-attention Vaswani et al. (2017). This allows adaptive weighting of neighbors during message passing, enhancing model expressivity and suitability for irregular graph structures, as detailed in recent surveys.

Efforts to adapt the Transformer architecture, initially designed for sequences, to graph data aim to capture long-range dependencies via global self-attention, potentially overcoming limitations of traditional message passing Ying et al. (2021), Yun et al. (2020). These Graph Transformers, reviewed systematically Shehzad et al. (2024), can incorporate positional information, handle diverse graph scales, achieve competitive results, and leverage language model pretraining benefits on text-rich graphs.

**Masked Pretraining Strategies**, popularized by language models Devlin et al. (2019) that learn contextual representations via masked token prediction, have been adapted for graphs. The Masked Graph Autoencoder (MaskGAE) Li et al. (2023) framework, for instance, masks graph edges and trains the model to reconstruct them, thereby learning robust representations for downstream tasks like link prediction and node classification.

**Integration of Language Models and Graph Structures** seeks to combine textual and structural information. Key directions include: fusing pretrained language models (LMs) with GNNs for joint reasoning over text and structure Plenz & Frank (2024); injecting structural information directly into LMs using graph-guided attention mechanisms, potentially removing the need for separate GNN modules Yuan & Färber (2024); interpreting Transformers as GNNs operating on complete graphs, allowing explicit modeling of edge information Henderson et al. (2023); and jointly training large LMs and GNNs to leverage both contextual understanding and structural learning capabilities Ioannidis et al. (2022).

## 3 Methodology

Given an attributed graph $G = (V, E)$ with node feature matrix $X \in \mathbb{R}^{|V| \times d}$ and binary adjacency matrix $A \in \{0, 1\}^{|V| \times |V|}$, our goal is to learn robust node representations that capture both the graph structure and semantic context. Each node $v_i \in V$ is associated with a feature vector $x_i \in \mathbb{R}^d$ and belongs to one of $C$ classes. To achieve this, we introduce (i) a Semantic Node Masking strategy that selectively perturbs node features based on structural importance, and (ii) a Dual-Branch Architecture that integrates graph-based and language-based embeddings.

### 3.1 Semantic Node Masking

Conventional random masking treats all nodes uniformly, often masking nodes that are crucial for maintaining graph connectivity. In contrast, our semantic masking strategy uses degree centrality to determine the importance of each node. By doing so, we ensure that nodes with higher connectivity, which are more influential in maintaining the overall structure are masked in a controlled manner. This design is inspired by graph theory concepts related to network resilience and influence maximization, where highly connected nodes play a pivotal role. Algorithm 1 provides an overview of the first step as seen in Fig. 1.

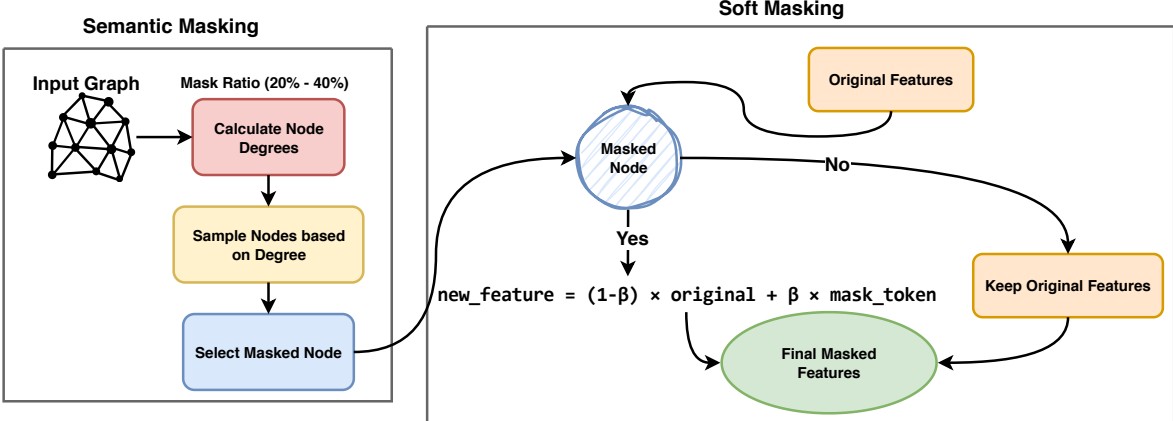

Figure 1: Semantic and Soft Masking of Nodes allows partial information to flow instead of binary on/off masking, the model can learn from both masked and original information simultaneously, creating a smoother, more continuous learning space with improved gradient flow.

---

**Algorithm 1** Semantic Node Mask Generation

---

**Require:** Graph data with:

- Node features $X \in \mathbb{R}^{N \times d}$
- Edge index $E$
- Boolean training mask `train_mask` (size $N$)
- Mask ratio $r \in (0, 1)$

**Ensure:** Boolean mask vector `mask` of length $N$
1: $train\_idx \leftarrow \{i \mid \texttt{train\_mask}[i] = \text{True}\}$
2: $num\_train \leftarrow |train\_idx|$
3: $num\_mask \leftarrow \max(1, \lfloor r \cdot num\_train \rfloor)$
4: **for** each node $i \in train\_idx$ **do**
5:     Compute degree: $d_i \leftarrow \sum_{j \in \mathcal{N}(i)} 1$
6: **end for**
7: Compute total degree: $D \leftarrow \sum_{i \in train\_idx} d_i$
8: **for** each node $i \in train\_idx$ **do**
9:     Set sampling probability: $p_i \leftarrow \frac{d_i}{D}$
10: **end for**
11: Sample **without replacement** a set $S \subseteq train\_idx$ of $num\_mask$ nodes according to probabilities $\{p_i\}$
12: Initialize `mask` as a Boolean vector of length $N$ with all entries `False`
13: **for** each node $i \in S$ **do**
14:     Set `mask[i]` $\leftarrow$ `True`
15: **end for**
16: **return** `mask`

---

### 3.1.1 Degree-based Node Selection

We compute the degree $d_i$ of a node $v_i$ as:

$$d_i = \sum_{j \in V} A_{ij}.$$

The probability of selecting a node for masking is proportional to its degree, computed over the training set $V_{\text{train}}$:

$$P(v_i) = \frac{d_i}{\sum_{j \in V_{\text{train}}} d_j}.$$

This selection strategy ensures that:

- Highly connected nodes, which are crucial for graph connectivity, are more likely to be masked.

- The masking probability distribution is adaptive to the topology of the graph.

- Only training nodes are considered for masking, thereby preserving the integrity of validation and test sets.

### 3.2   Soft Masking Mechanism

Algorithm 2 provides an overview of the second step, as illustrated in Figure 1. Unlike binary masking approaches that completely remove a node's feature information, we propose a *soft masking* strategy that interpolates between the original feature vector and a learnable mask token $m \in \mathbb{R}^d$. This design ensures that critical information is not abruptly removed, improving gradient flow and optimizing information retention.

Specifically, for a node $v_i$, the masked feature $\tilde{x}_i$ is computed as follows:

$$\tilde{x}_i = \begin{cases} (1 - \beta)x_i + \beta\, m, & \text{if } v_i \text{ is selected for masking,} \\ x_i, & \text{otherwise.} \end{cases} \tag{1}$$

The interpolation coefficient $\beta \in [0, 1]$ controls the degree of masking. In our experiments, we set $\beta = 0.7$, striking a balance between preserving original node features and introducing masked noise. This gradual transformation allows partial information to persist, leading to:

- **Improved Gradient Flow:** Unlike hard masking, which creates sharp discontinuities in the loss landscape, soft masking ensures a smoother optimization process. This continuous transition reduces gradient variance and mitigates training instability.

- **Enhanced Information Retention:** Soft masking maintains a partial connection between masked nodes and their original features, preserving important semantic and structural information.

- **Alignment with Information Theory Concepts:** Inspired by the information bottleneck principle Tishby et al. (2000), soft masking ensures that only a controlled amount of information is removed, reducing redundancy while preserving essential task-relevant features.

### 3.3   Dual-Branch Architecture

The novelty of GMLM also lies in its dual-branch design that integrates a GAT branch with a BERT branch. The GAT branch captures the intrinsic graph structure using attention mechanisms, while the BERT branch extracts rich semantic representations from node-associated text. A multi-layer fusion network then combines these embeddings to produce a robust node representation that benefits from both structural and semantic perspectives. This synergistic combination overcomes the limitations seen in methods that rely exclusively on one modality.

#### 3.3.1   Graph Attention Branch

Our model consists of two branches that extract complementary features from the graph. The first GAT layer computes hidden representations as:

$$h_i^{(1)} = \text{ELU}\Big(\text{BN}\Big(\Big\|_{k=1}^{K_1} \sum_{j \in \mathcal{N}_i} \alpha_{ij,k}^{(1)} W_k^{(1)} \tilde{x}_j\Big)\Big),$$

---

**Algorithm 2** Soft Masking of Node Features

---

**Require:**
  - Node feature matrix $X \in \mathbb{R}^{N \times d}$
  - Boolean mask vector `mask` (length $N$)
  - Learnable mask token $m \in \mathbb{R}^d$
  - Interpolation coefficient $\beta \in [0, 1]$

**Ensure:** Soft-masked feature matrix $\tilde{X} \in \mathbb{R}^{N \times d}$

1: $\tilde{X} \leftarrow X$                                                            ▷ Copy original features
2: **for** each node $i = 1$ to $N$ **do**
3:     **if** `mask[i]` is **True then**
4:         Update feature: $\tilde{X}[i] \leftarrow (1 - \beta) \cdot X[i] + \beta \cdot m$
5:     **end if**
6: **end for**
7: **return** $\tilde{X}$

---

where:

- $K_1 = 4$ is the number of attention heads.

- $\mathcal{N}_i$ denotes the set of neighbors of node $v_i$.

- $W_k^{(1)} \in \mathbb{R}^{d' \times d}$ is the learnable weight matrix for head $k$.

- $\alpha_{ij,k}^{(1)}$ are attention coefficients computed by:

$$\alpha_{ij,k}^{(1)} = \frac{\exp\big(\text{LeakyReLU}(a_k^T [W_k^{(1)} \tilde{x}_i \,\|\, W_k^{(1)} \tilde{x}_j])\big)}{\sum_{l \in \mathcal{N}_i} \exp\big(\text{LeakyReLU}(a_k^T [W_k^{(1)} \tilde{x}_i \,\|\, W_k^{(1)} \tilde{x}_l])\big)},$$

  with $a_k \in \mathbb{R}^{2d'}$ being the attention vector for head $k$, and $\|$ denoting concatenation.

- BN and ELU denote batch normalization and the Exponential Linear Unit activation function, respectively.

The second GAT layer refines the representations as:

$$h_i^{(2)} = \text{ELU}\Big(\text{BN}\Big(\big\|_{k=1}^{K_2} \sum_{j \in \mathcal{N}_i} \alpha_{ij,k}^{(2)} W_k^{(2)} h_j^{(1)}\big)\Big)\Big),$$

where $K_2 = 2$ denotes the number of attention heads in the second layer.

### 3.3.2 BERT Branch

We use DistilBERT Sanh et al. (2020) as our masking model. The BERT branch processes node states as tokens:

$$t_i = \begin{cases} \text{"[MASK]"}, & \text{if } v_i \text{ is masked,} \\ \text{"node"}, & \text{otherwise.} \end{cases}$$

The token is then processed by BERT:

$$b_i = \text{BERT}(\text{tokenize}(t_i)),$$

yielding the BERT embedding $b_i \in \mathbb{R}^{d_b}$ for node $v_i$. These embeddings capture contextual information that complements the structural cues obtained from the GAT branch.

**Determinism at Inference**  During testing, DistilBERT runs with frozen weights and inference-mode (dropout disabled), so identical tokens (`"node"` or `"[MASK]"`) always yield identical embeddings.

**Why a PLM instead of Static Vectors?**  We fine-tune a PLM rather than learn two random vectors because:

1. **Rich Initialization** Embeddings inherit semantic priors from large-scale pre-training.

2. **Task Adaptation** Fine-tuning aligns them for optimal fusion with GAT structural features.

3. **Higher Capacity** The transformer architecture models complex interactions that fixed vectors cannot.

### 3.3.3  Multi-layer Fusion Network

The fusion network combines the graph structural features and the BERT contextual embeddings. First, the two embeddings are concatenated:

$$z_i = \left[ h_i^{(2)} \,\|\, b_i \right].$$

Next, the concatenated vector is passed through a two-layer fully-connected network with residual connections and layer normalization:

$$f_i = \text{LayerNorm}\Big( W_2 \,\text{ReLU}\big(\text{LayerNorm}(W_1 z_i)\big) \Big),$$

where $W_1$ and $W_2$ are learnable weight matrices. This fusion step ensures that both structural and semantic features are effectively integrated while maintaining stable training dynamics.

### 3.4  Node Classification

The final node representations $f_i$ are used to perform node classification. A classification head maps the fused representations to class probabilities:

$$\hat{y}_i = \text{softmax}\Big( W_c \,\text{ReLU}(W_h f_i) \Big),$$

where $W_h$ and $W_c$ are learnable projection matrices. The softmax function ensures that the output $\hat{y}_i \in \mathbb{R}^C$ represents a valid probability distribution over the $C$ classes.

## 4  Training

All experiments and evaluations were run using a singular V100(32GB). The average VRAM usage was between  22GB, with a peak usage at 30GB VRAM usage.

### 4.1  Node Classification & Graph Classification

To train GMLM for node classification tasks we follow the below described procedure. We also provide dataset details such as node and edge counts in Appendix D.1.

Our training framework consists of two phases:

1. **Contrastive Pretraining:** Learn robust node representations by aligning embeddings from different views of the graph.

2. **Supervised Fine-Tuning:** Adapt the pretrained representations for node classification using a label-smoothed cross-entropy loss.

### 4.1.1  Contrastive Pretraining

The goal of the contrastive pretraining phase is to maximize the agreement between different views of the same node while ensuring separation from other nodes.

### 4.1.2  View Generation

For each training iteration, two views of the graph are generated by applying distinct random masks to the node features:
$$\tilde{X}_1 = \text{Mask}(X, r_1), \quad \tilde{X}_2 = \text{Mask}(X, r_2),$$
where the mask ratios $r_1$ and $r_2$ are independently sampled from:

$$r_1, r_2 \sim \text{Uniform}(0.2, 0.4).$$

This procedure provides complementary views that challenge the model to learn invariant representations.

### 4.1.3  Contrastive Loss

We adopt the NT-Xent (Normalized Temperature-scaled Cross Entropy) loss Sohn (2016) to align the embeddings from the two views. Let $z_i$ and $z_i'$ denote the $\ell_2$-normalized embeddings for node $v_i$ from the two views. The loss is defined as:

$$\mathcal{L}_{\text{cont}} = \frac{1}{2N} \sum_{i=1}^{N} \left[ -\log \frac{\exp\left(z_i^T z_i'/\tau\right)}{\sum_{k=1}^{2N} \mathbb{1}_{[k \neq i]} \exp\left(z_i^T z_k/\tau\right)} \right],$$

where:

- $\tau = 0.5$ is the temperature parameter.

- $N$ is the number of nodes.

- $\mathbb{1}_{[k \neq i]}$ is an indicator function that excludes the positive pair from the denominator.

### 4.1.4  Supervised Fine-Tuning

In the fine-tuning phase, the pretrained model is optimized for the node classification task.

### 4.1.5  Loss Function with Label Smoothing

We utilize a cross-entropy loss with label smoothing to prevent overconfidence. The loss function is given by:

$$\mathcal{L}_{\text{cls}} = - \sum_{i \in V_{\text{train}}} \left[ (1 - \epsilon)\, y_i \log(\hat{y}_i) + \epsilon \sum_{j \neq y_i} \frac{1}{C-1} \log(\hat{y}_j) \right],$$

where:

- $\epsilon = 0.1$ is the smoothing parameter.

- $y_i$ denotes the ground-truth label (expressed in one-hot encoding).

- $\hat{y}_i$ is the predicted probability for the correct class.

- $C$ is the number of classes.

### 4.1.6  Learning Rate Strategy

Different components of the model are optimized with different learning rates:

$$\eta = \begin{cases} 10^{-3} & \text{for graph parameters,} \\ 10^{-5} & \text{for BERT parameters,} \\ 10^{-4} & \text{for other parameters.} \end{cases}$$

We employ cosine annealing with warm restarts to update the learning rate during training:

$$\eta_t = \eta_{\min} + \frac{1}{2}(\eta_{\max} - \eta_{\min})\left(1 + \cos\left(\frac{T_{\mathrm{cur}}}{T_i}\pi\right)\right),$$

where:

- $T_{\mathrm{cur}}$ is the number of epochs since the last restart.

- $T_i$ is the total number of epochs in the current cycle.

- The initial cycle is set to $T_0 = 20$ epochs.

- A multiplication factor $T_{\mathrm{mult}} = 2$ is used to extend each subsequent cycle.

### 4.2 Language understanding

While GLUE tasks do not provide explicit graph structures, we construct a token-level graph for each input sequence. Specifically, given an input sentence or sentence pair tokenized as $x = [x_1, x_2, \ldots, x_n]$, we define a fully connected undirected graph $G = (V, E)$, where $V = \{x_1, x_2, \ldots, x_n\}$ and $E = \{(x_i, x_j) \mid 1 \le i, j \le n\}$.

Each node $x_i$ is initialized with its corresponding token embedding, and edges are used to compute pairwise attention scores, similar to standard self-attention but within a graph neural network framework. This allows GMLM to process GLUE inputs by treating them as dense graphs over token embeddings, enabling the model to capture contextual relationships.

To train **GMLM** for language understanding tasks, we follow a two-phase setup similar to node classification. However, instead of using fixed hyperparameters, we employ *grid search* for hyperparameter tuning. We also provide dataset details and their evaluation metrics in Appendix D.2.

## 5 Results

### 5.1 Node Classification

Table 1 summarizes the test accuracy and standard deviation across six benchmarks. Our Graph Masked Language Model (GMLM) is compared against self-supervised methods (*BGRL* Sohn (2016), *S3-CL* Ding et al. (2022), *GraphMAE* Hou et al. (2022)) and semi/unsupervised baselines (*GCN* Kipf & Welling (2017), *SUGRL* Mo et al. (2022)). Results follow the evaluation protocols in Veličković et al. (2018); Ju et al. (2023); Ding et al. (2022), averaged over 7 runs.

GMLM achieves SOTA performance on multiple datasets: **87.5%** on **Cora**, **75.7%** on **Citeseer**, and **86.2%** on **Pubmed** Yang et al. (2016). On **Coauthor CS** and **Amazon Photo** Shchur et al. (2019), GMLM reaches **93.7%** and **93.6%**, respectively.

Ablation results show that *GMLM w/o Semantic Masking* maintains high mean accuracy but suffers from higher standard deviations, notably **1.1** on Citeseer and **6.8** on Amazon Computers. *GMLM w/o Soft Masking* degrades significantly, particularly on Amazon Computers (**71.8 ± 11.6**).

The full GMLM, combining both masking strategies, consistently delivers the highest accuracy and stability.

### 5.2 Graph Classification

Table 2 reports the graph classification accuracies across four benchmark datasets from TUDatasets Morris et al. (2020). GMLM shows strong performance on social network datasets such as COLLAB and IMDB-B, achieving the highest accuracy on IMDB-B and third best on COLLAB. However, on bioinformatics datasets like PROTEINS and MUTAG, GMLM trails behind leading methods. These results underline GMLM's effectiveness in capturing the structure of social graphs, while highlighting limitations in modeling more complex biological topologies.

| Method | Cora | Citeseer | Pubmed | Amazon P | Amazon Comp. | Coauthor CS |
|---|---|---|---|---|---|---|
| DGI | 81.7 ± 0.6 | 71.5 ± 0.7 | 77.3 ± 0.6 | 83.1 ± 0.3 | 83.6 ± 0.2 | 90.0 ± 0.3 |
| MVGRL | 82.9 ± 0.7 | 72.6 ± 0.7 | 79.4 ± 0.3 | 87.3 ± 0.1 | 82.8 ± 0.1 | 91.3 ± 0.1 |
| GRACE | 80.0 ± 0.4 | 71.7 ± 0.6 | 79.5 ± 1.1 | 81.8 ± 0.8 | 89.5 ± 0.3 | 71.1 ± 0.2 |
| CCA-SSG | 84.2 ± 0.4 | 73.1 ± 0.3 | 81.6 ± 0.4 | 93.1 ± 0.1 | 88.7 ± 0.3 | 93.3 ± 0.2 |
| SUGRL | 83.4 ± 0.5 | 73.0 ± 0.4 | 81.9 ± 0.3 | 93.2 ± 0.4 | 88.9 ± 0.2 | 92.2 ± 0.5 |
| S3-CL | 84.5 ± 0.4 | 74.6 ± 0.4 | 80.8 ± 0.3 | 89.0 ± 0.5 | N/A | 93.1 ± 0.4 |
| GraphMAE | 84.2 ± 0.4 | 73.1 ± 0.4 | 83.9 ± 0.3 | 90.7 ± 0.4 | 79.4 ± 0.5 | 93.1 ± 0.1 |
| GraphMAE2 | 84.4 ± 0.5 | 73.4 ± 0.3 | 81.4 ± 0.5 | N/A | N/A | N/A |
| UGMAE | 85.1 ± 0.4 | 73.9 ± 0.3 | 82.2 ± 0.1 | N/A | N/A | N/A |
| GMI | 82.7 ± 0.2 | 73.3 ± 0.3 | 77.3 ± 0.6 | 83.1 ± 0.3 | N/A | 91.0 ± 0.0 |
| BGRL | 83.8 ± 1.6 | 72.3 ± 0.9 | 86.0 ± 0.3 | 93.2 ± 0.3 | **90.3 ± 0.2** | 93.3 ± 0.1 |
| GCN | 81.5 ± 1.3 | 71.9 ± 1.9 | 77.8 ± 2.9 | 91.2 ± 1.2 | 82.6 ± 2.4 | 91.1 ± 0.5 |
| GAT | 81.8 ± 1.3 | 71.4 ± 1.9 | 78.7 ± 2.3 | 85.7 ± 20.3 | 78.0 ± 19.0 | 90.5 ± 0.6 |
| MoNet | 81.3 ± 1.3 | 71.2 ± 2.0 | 78.6 ± 2.3 | 91.2 ± 1.3 | 83.5 ± 2.2 | 90.8 ± 0.6 |
| GS-mean | 79.2 ± 7.7 | 71.6 ± 1.9 | 77.4 ± 2.2 | 91.4 ± 1.3 | 82.4 ± 1.8 | 91.3 ± 2.8 |
| GS-maxpool | 76.6 ± 1.9 | 67.5 ± 2.3 | 76.1 ± 2.3 | 90.4 ± 1.3 | N/A | 85.0 ± 1.1 |
| GS-meanpool | 77.9 ± 2.4 | 68.6 ± 2.4 | 76.5 ± 2.4 | 90.7 ± 1.6 | 79.9 ± 2.3 | 89.6 ± 0.9 |
| MLP | 58.2 ± 2.1 | 59.1 ± 2.3 | 70.0 ± 2.1 | 69.6 ± 3.8 | 44.9 ± 5.8 | 88.3 ± 0.7 |
| LogReg | 57.1 ± 2.3 | 61.0 ± 2.2 | 64.1 ± 3.1 | 73.0 ± 6.5 | 64.1 ± 5.7 | 86.4 ± 0.9 |
| LabelProp | 74.4 ± 2.6 | 67.8 ± 2.1 | 70.5 ± 5.3 | 72.6 ± 11.1 | 70.8 ± 8.1 | 73.6 ± 3.9 |
| LabelProp NL | 73.9 ± 1.6 | 66.7 ± 2.2 | 72.3 ± 2.9 | 83.9 ± 2.7 | 75.0 ± 2.9 | 76.7 ± 1.4 |
| GMLM (w/o Semantic Masking) | 86.8 ± 0.6 | 75.6 ± 1.1 | 85.3 ± 0.3 | 92.9 ± 0.4 | 78.3 ± 6.8 | 93.0 ± 0.3 |
| GMLM (w/o Soft Masking) | 85.2 ± 1.8 | 74.8 ± 1.1 | 85.4 ± 0.1 | 93.5 ± 0.8 | 71.8 ± 11.6 | 92.4 ± 0.6 |
| GMLM | **87.5 ± 0.9** | **75.7±0.6** | **86.2±0.2** | **93.6±0.5** | 85.1±1.7 | **93.7±0.3** |

Table 1: Mean test set accuracy and standard deviation on Node Classification tasks across 7 runs in percent. Highest scores are in **bold**, second highest are underlined.

### 5.3 Link Prediction

In Table 3 we see the performance of GMLM on link prediction tasks (CORA, CiteSeer and PubMed). We clearly see it having comparable performance with GAE, VGAEKipf & Welling (2016) with the baselines, giving the best performance on Cora and only being 0.2% off on PubMed when compared to GAE.

### 5.4 Language Understanding

Through extensive grid search, we identified the best-performing hyperparameters for each dataset in the GLUE benchmark Wang et al. (2019). We optimized three key variables: `temperature`, `pre_train_epochs`, and `finetune_epochs`. The best-performing settings for each task are summarized in Table 7.

Table 4 shows the performance on the following tasks CoLA, MRPC, RTE, SST-2, STS-B, and WNLI. We compare GMLM against well-established models such as **ELMo** Peters et al. (2018), **BERT-base** Devlin et al. (2019), and **DistilBERT** Sanh et al. (2020).

GMLM, using DistilBERT as its masking model, demonstrates a performance improvement of **1.1** points on average compared to the baseline DistilBERT. Notably, GMLM achieves the best results on MRPC and WNLI, surpassing even the performance of BERT-base.

## 6 Ablation Studies

We present two-dimensional embeddings of the node representations learned by our Graph Masked Language Model (GMLM). We use both t-SNE and UMAP to project the high-dimensional embeddings onto two dimensions. CITESEER contain nodes (papers) from different research areas, labeled as `Agents`, `AI`, `DB`, `IR`, `ML`, and `HCI`.

| Model | COLLAB | IMDB-B | PROTEINS | MUTAG |
|---|---|---|---|---|
| GK | 72.84 ± 0.28 | 65.87 ± 0.98 | 71.67 ± 0.55 | 81.58 ± 2.11 |
| WL | 79.02 ± 1.77 | 73.40 ± 4.63 | 74.68 ± 0.49 | 82.05 ± 0.36 |
| PSCN | 72.60 ± 2.15 | 71.00 ± 2.29 | 75.89 ± 2.76 | **92.63 ± 4.21** |
| GCN | **81.72 ± 1.64** | 73.30 ± 5.29 | 75.65 ± 3.24 | 87.20 ± 5.11 |
| GFN | 81.50 ± 2.42 | 73.00 ± 4.35 | 76.46 ± 4.06 | 90.84 ± 7.22 |
| GraphSAGE | 79.70 ± 1.70 | 72.40 ± 3.60 | 65.90 ± 2.70 | 79.80 ± 13.9 |
| GAT | 75.80 ± 1.60 | 70.50 ± 2.30 | 74.70 ± 2.20 | 89.40 ± 6.10 |
| DGCNN | 73.76 ± 0.49 | 70.03 ± 0.86 | 75.54 ± 0.94 | 85.83 ± 1.66 |
| PPGN | 81.38 ± 1.42 | 73.00 ± 5.77 | 77.20 ± 4.73 | 90.55 ± 8.70 |
| CapsGNN | 79.62 ± 0.91 | 73.10 ± 4.83 | 76.28 ± 3.63 | 86.67 ± 6.88 |
| DSGC | 79.20 ± 1.60 | 73.20 ± 4.90 | 74.20 ± 3.80 | 86.70 ± 7.60 |
| GIN-0 | 80.20 ± 1.90 | 75.10 ± 5.10 | 76.20 ± 2.80 | 89.40 ± 5.60 |
| IEGN | 77.92 ± 1.70 | 71.27 ± 4.50 | 75.19 ± 4.30 | 84.61 ± 10.0 |
| U2GNN | 77.84 ± 1.48 | 77.00 ± 3.45 | 78.53 ± 4.07 | 89.97 ± 3.65 |
| GraphMAE | 75.52 ± 0.66 | 75.30 ± 0.39 | 80.32 ± 0.46 | 88.19 ± 1.26 |
| GraphMAE2 | 73.88 ± 0.53 | 74.86 ± 0.34 | 77.59 ± 0.22 | 86.63 ± 1.33 |
| UGMAE | 76.06 ± 0.59 | 76.78 ± 0.22 | **81.66 ± 0.12** | 88.26 ± 1.19 |
| GMLM (w/o Semantic Masking) | 78.12 ± 1.27 | 75.64 ± 1.93 | 68.45 ± 5.31 | 79.49 ± 3.58 |
| GMLM (w/o Soft Masking) | 74.36 ± 2.01 | 74.84 ± 3.58 | 65.37 ± 5.70 | 77.98 ± 4.57 |
| GMLM | 80.00 ± 0.93 | **77.00 ± 1.73** | 71.61 ± 3.14 | 82.80 ± 4.20 |

Table 2: Mean test set accuracy and standard deviation on Graph Classification tasks using standard practices as described in TUDatasets. Highest scores are denoted by **bold**.

| Method | Cora | | Citeseer | | Pubmed | |
|---|---|---|---|---|---|---|
| | AUC | AP | AUC | AP | AUC | AP |
| SC | 84.6±0.01 | 88.5±0.00 | 80.5±0.01 | 85.0±0.01 | 84.2±0.02 | 87.8±0.01 |
| DW | 83.1±0.01 | 85.0±0.00 | 80.5±0.02 | 83.6±0.01 | 84.4±0.01 | 84.1±0.00 |
| GAE* | 84.3±0.02 | 88.1±0.01 | 78.7±0.02 | 84.1±0.02 | 82.2±0.01 | 87.4±0.00 |
| VGAE* | 84.0±0.02 | 87.7±0.01 | 78.9±0.03 | 84.1±0.02 | 82.7±0.01 | 85.7±0.01 |
| GAE | 91.0±0.02 | 92.0±0.03 | 89.5±0.04 | 89.9±0.05 | **96.4±0.00** | **96.5±0.00** |
| VGAE | 91.4±0.01 | **92.6±0.01** | **90.8±0.02** | **92.0±0.02** | 94.4±0.02 | 94.7±0.02 |
| GMLM | **92.1±0.00** | 92.4±0.00 | 87.3±0.01 | 89.2±0.01 | 96.2±0.00 | 96.3±0.00 |

Table 3: Mean test set accuracy and standard deviation on link prediction tasks. * indicates models without input features. **Bold** denotes the best performing method.

| Model | Score | CoLA | MRPC | RTE | SST-2 | STS-B | WNLI |
|---|---|---|---|---|---|---|---|
| ELMo | 66.6 | 44.1 | 76.6 | 53.4 | 91.5 | 70.4 | 56.3 |
| BERT-base | **74.3** | **56.3** | 88.6 | **69.3** | **92.7** | **89.0** | 53.5 |
| DistilBERT | 72.2 | 51.3 | 87.5 | 59.9 | 91.3 | 86.9 | 56.3 |
| GMLM (w/o Semantic Masking) | 71.6 | 51.5 | 86.8 | 59.8 | 88.4 | 85.9 | 57.7 |
| GMLM (w/o Soft Masking) | 70.6 | 49.7 | 86.4 | 57.9 | 87.6 | 84.8 | 57.2 |
| GMLM | 73.3 | 52.7 | **88.8** | 61.3 | 89.6 | 87.0 | **60.6** |

Table 4: Evaluation on GLUE. **Bold** denotes the best score, and underline denotes the second best. ELMo, BERT, and DistilBERT results are as reported by the authors.

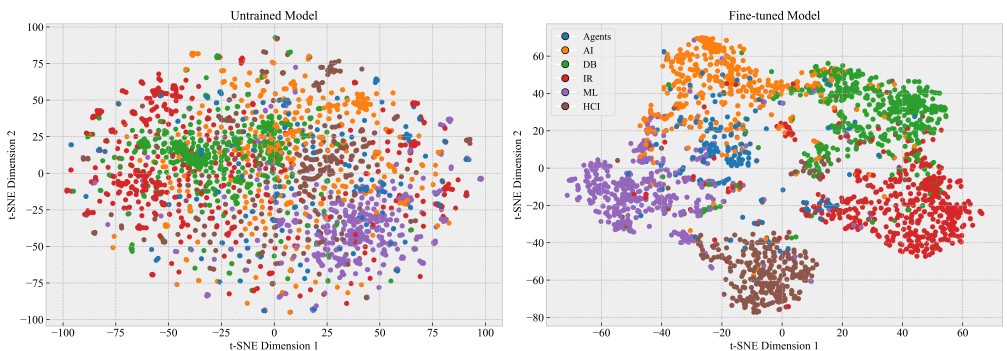

Figure 2: t-SNE embeddings (perplexity = 30) for the untrained (left) and fine-tuned (right) Graph Masked Language Model. Points are colored by research area: `Agents`, `AI`, `DB`, `IR`, `ML`, and `HCI`.

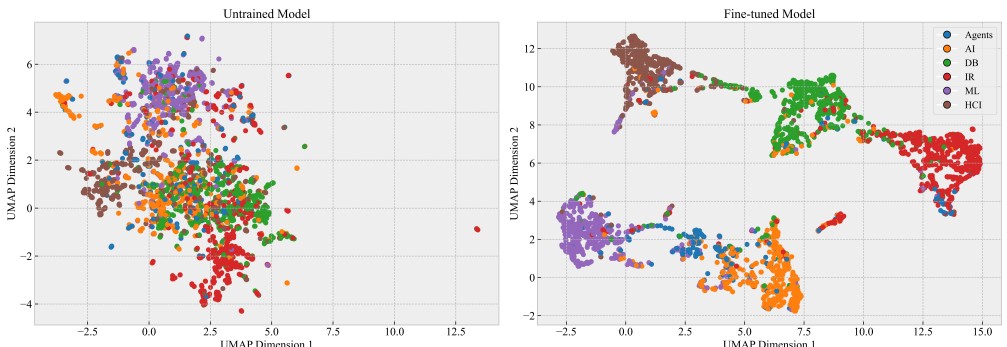

Figure 3: UMAP embeddings (`n_neighbors = 15`, `min_dist = 0.1`) for the untrained (left) and fine-tuned (right) Graph Masked Language Model. Each color corresponds to a different research area.

## 6.1  t-SNE Embeddings (Perplexity = 30)

Figure 2 shows t-SNE projections of GMLM node embeddings on CITESEER. Left: untrained model; Right: fine-tuned model. With perplexity 30, the untrained embeddings form diffuse groups, lacking clear initial structure. The fine-tuned model yields distinct clusters for research areas (`Agents`, `AI`, `DB`, `IR`, `ML`, `HCI`), showing improved representations.

Figure 4 compares t-SNE embeddings for GMLM variants: (**Left**) without Soft Masking, (**Center**) without Semantic Masking, (**Right**) complete GMLM. The full model produces the most compact and separate clusters, confirming the benefit of both masking strategies for finding data structure.

## 6.2  UMAP Embeddings (n_neighbors = 15, min_dist = 0.1)

Figure 3 displays UMAP embeddings (`n_neighbors=15`, `min_dist=0.1`). The untrained model (left) has weak spatial structure and high category overlap. The fine-tuned version (right) produces tight, separate clusters, showing the learning process embeds relevant data and distinguishes classes.

Figure 5 provides UMAP projections for the ablation study: (**Left**) without Soft Masking, (**Center**) without Semantic Masking, (**Right**) Full GMLM. The full model yields the clearest class divisions. This supports that both masking methods jointly improve representation quality. See Appendix A for masking parameter sensitivity analysis.

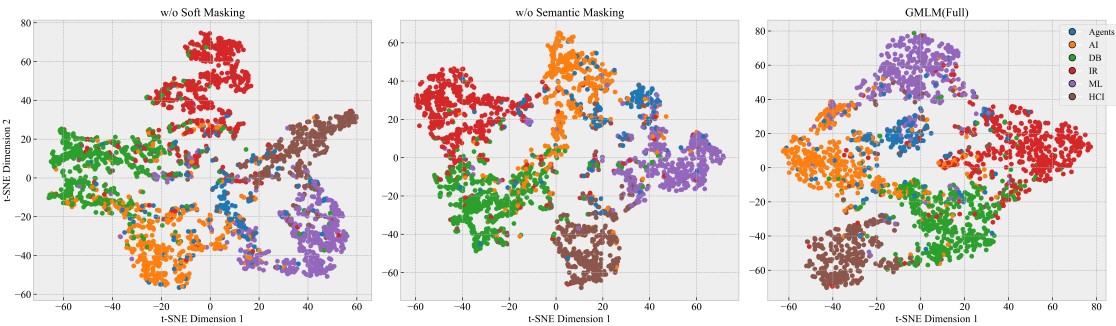

Figure 4: t-SNE visualization of embeddings for different ablated versions. (**Left**) w/o Soft Masking, (**Center**) w/o Semantic Masking, (**Right**) GMLM(Full).

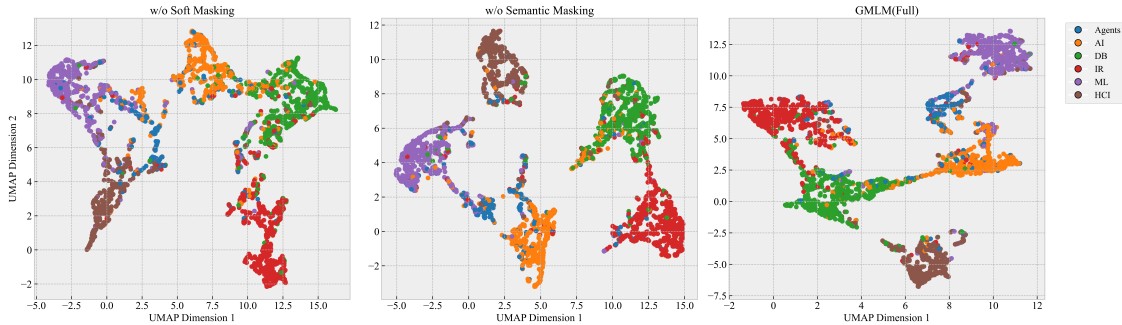

Figure 5: UMAP visualization of embeddings for different ablated versions. (**Left**) w/o Soft Masking, (**Center**) w/o Semantic Masking, (**Right**) GMLM(Full).

# 7 Conclusion

This paper introduces Graph Masked Language Models (GMLM) to enhance node and graph classifcicationin graph neural networks. GMLM utilizes a two-step masking strategy, integrating graph attention networks (GATs) with masking utilizing pre-trained language models like BERT. This approach improves feature learning via selective node attribute masking and reconstruction, leading to better generalization and robustness. Extensive experiments show GMLM achieves or is competitive with state-of-the-art (SOTA) on node and graph classification tasks. On language understanding tasks such as GLUE, GMLM surpasses comparison models on WNLI/MRPC, ranking second overall. Our results also highlight the importance of both Semantic and Soft Masking, which jointly contribute to more stable and reliable performance across datasets.

# 8 Limitations

While we achieve state-of-the-art performance using DistilBERT, we have not yet explored the use of larger masked language models. Our decision to use DistilBERT was driven by the need for efficiency, as larger models come with increased computational costs and longer training times. For this study, we prioritized a more lightweight yet effective approach to demonstrate that our framework can achieve competitive performance compared to other self-supervised, semi-supervised, and unsupervised training methods for node classification tasks. Future work could explore the impact of larger models on performance and assess the trade-off between efficiency and accuracy.

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

# A  Parameter Sweep for Citeseer

In this section, we present the parameter sweep results for the GMLM (Graph Masked Language Model) on the Citeseer dataset on **one run**. We visualize these results using parallel coordinate plots, exploring different masking strategies. Our analysis compares three distinct configurations to understand the impact of various masking approaches on model performance.

We examine:

1. **Full GMLM with semantic and soft masking** (Figure 6).

2. **GMLM with only semantic masking** (Figure 7).

3. **GMLM with only soft masking (no semantic masking)** (Figure 8).

Each figure utilizes a parallel coordinate plot to illustrate the relationship between hyperparameter variations and model accuracy. The color bar on the right of each plot visually encodes accuracy, ranging from 71% to 76%. Within the plots, lines represent individual parameter combinations. For visualization purposes, each parameter value is normalized along its respective vertical axis.

## A.1  Full GMLM with Semantic and Soft Masking

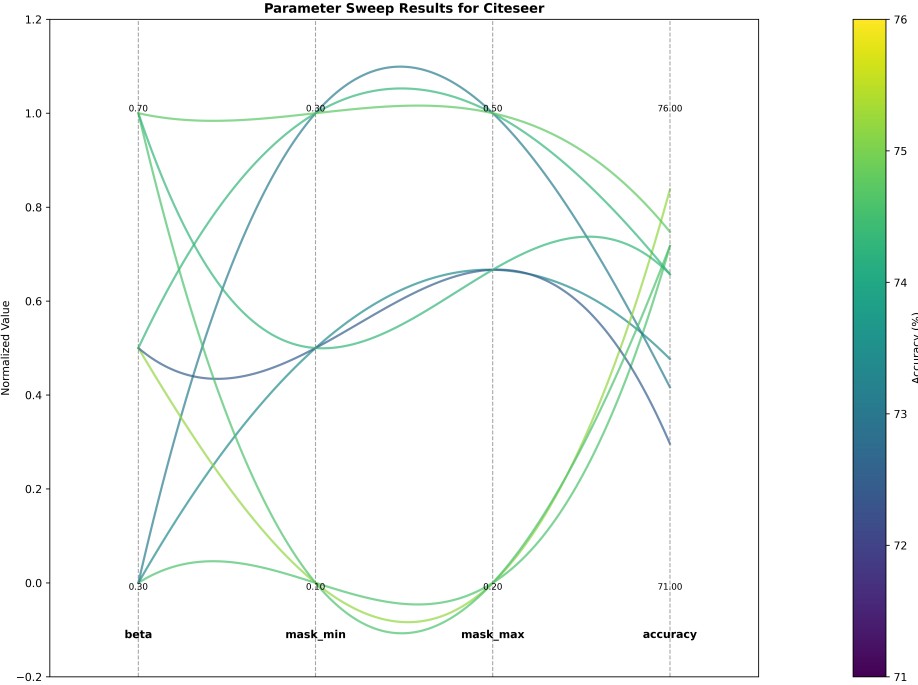

Figure 6: Parameter sweep results for the full GMLM with both semantic and soft masking. The parallel coordinates plot displays four axes: *beta*, *mask_min*, *mask_max*, and *accuracy*. Lines are colored according to model accuracy.

Figure 6 illustrates the parameter sweep results when both semantic and soft masking are active in the GMLM framework. The parameters explored in this configuration are *beta*, *mask_min*, and *mask_max*, alongside the resulting model accuracy. Specifically, *beta* (often denoted as $\beta$) is the weighting factor that governs the contribution of semantic masking to the overall masking strategy. The parameters *mask_min* and *mask_max* define the minimum and maximum thresholds for applying masking, respectively. Finally, *accuracy* represents the model's performance on the Citeseer dataset.

Observing Figure 6, we can discern several trends. When the value of $\beta$ is excessively low, the influence of semantic masking may be insufficient. This can hinder the model's ability to focus on semantically relevant context, potentially leading to reduced accuracy. In contrast, a high $\beta$ might cause the model to overemphasize semantic relationships. This over-reliance could lead to the model neglecting other informative patterns that soft masking is designed to capture, also resulting in suboptimal performance.

Furthermore, *mask_min* and *mask_max* play a crucial role in determining the aggressiveness of the masking process. These parameters directly influence how many tokens (or nodes/edges in a graph context) are masked during training. The level of masking difficulty, therefore, is modulated by these thresholds. The results suggest that a moderate range for both *mask_min* and *mask_max*, when combined with a balanced $\beta$ value, tends to yield the most favorable outcomes. This is evidenced by the higher accuracy values, often in the 75–76% range, observed in these regions of the parameter space.

**Importance of Combined Semantic and Soft Masking.** The effectiveness of the full GMLM configuration stems from the synergistic interaction between semantic and soft masking. Semantic masking is designed to prioritize the masking of tokens that are deemed more "meaningful" based on semantic relationships within the graph. Soft masking, in contrast, provides a complementary approach by capturing more generalizable and structural patterns present in the data.

By integrating both strategies, we achieve a more comprehensive masking approach. This ensures that the model is exposed to a diverse set of masked positions during training. These positions include both semantically salient elements and those capturing broader structural variations. A well-tuned $\beta$ parameter is essential to balance these two masking aspects, preventing the model from becoming overly biased towards a single masking strategy. In conclusion, achieving an optimal balance between $\beta$ and the masking ratio (defined by *mask_min* and *mask_max*) is key. This balance facilitates the learning of rich and robust representations, ultimately manifesting in improved model accuracy.

## A.2 GMLM with Only Semantic Masking

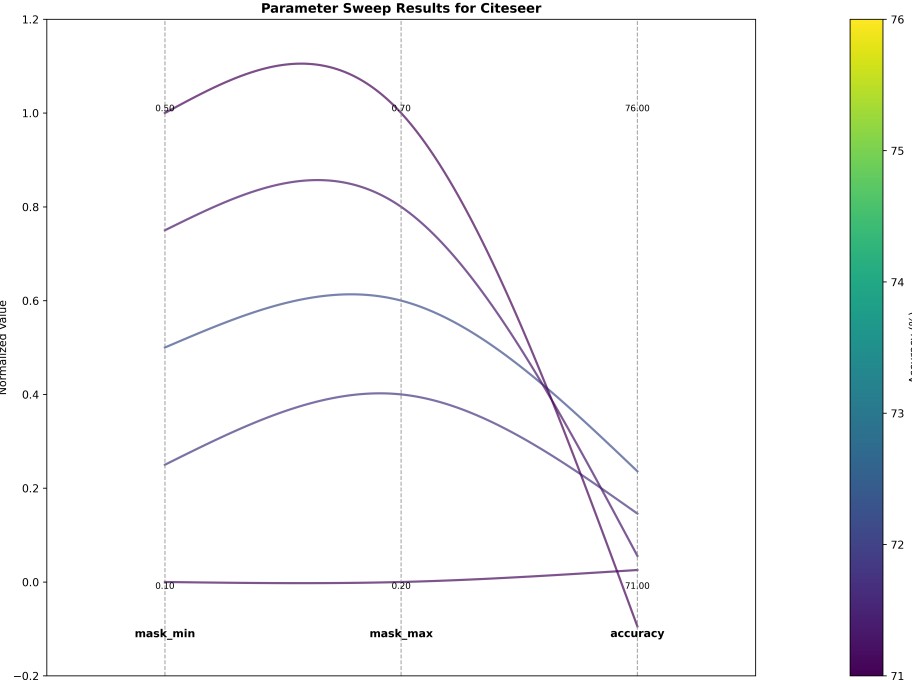

Figure 7: Parameter sweep results for GMLM with **only** semantic masking. The axes are *mask_min*, *mask_max*, and *accuracy*.

Figure 7 presents the parameter sweep results for a GMLM configuration that employs only semantic masking. In this scenario, soft masking is disabled, and consequently, the *beta* parameter is not relevant and therefore absent from the figure. The parameters under investigation are solely *mask_min* and *mask_max*, along with the resultant *accuracy*.

The plot reveals how variations in *mask_min* and *mask_max* influence the intensity of semantic masking. This, in turn, affects the difficulty of the masked language model prediction task. Notably, the highest accuracy values are still concentrated within a specific region of the *mask_min–mask_max* space. This observation suggests that both excessively aggressive and overly lenient masking ratios can negatively impact model performance, even when relying solely on semantic masking.

**Effect of Semantic-Only Masking.** Utilizing semantic information in isolation can be advantageous when the underlying dataset structure strongly aligns with semantic relationships. In such cases, focusing on semantic masking may effectively guide the model. However, this approach also carries the risk of overfitting to specific semantic patterns present in the training data. Without the complementary influence of soft masking, the model's ability to generalize to broader structural variations within the graph may be compromised.

## A.3   GMLM with Only Soft Masking

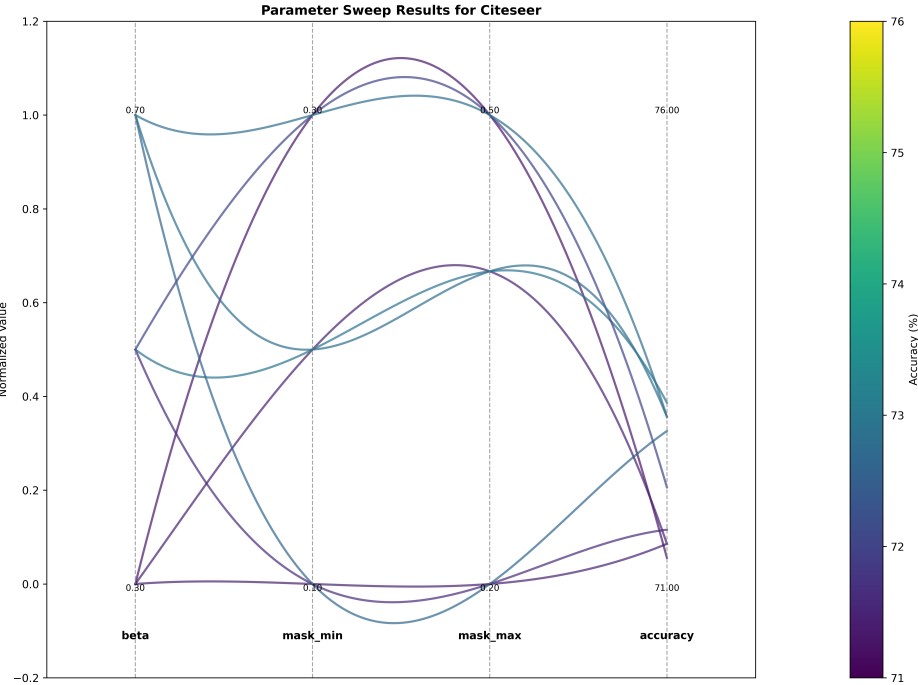

Figure 8: Parameter sweep results for GMLM with **only** soft masking (no semantic masking). The axes are *mask_min*, *mask_max*, and *accuracy*.

Figure 8 displays the results of the parameter sweep when semantic masking is deactivated and only soft masking is used. Similar to the semantic-only configuration, the parameters under consideration are *mask_min*, *mask_max*, and the resulting *accuracy*.

Analyzing this plot, we observe a similar trend to the semantic-only setup. The careful tuning of *mask_min* and *mask_max* remains crucial for achieving competitive accuracy levels. The performance of the model is directly related to the extent of soft masking applied. While soft masking can effectively highlight more generalizable features within the data, it lacks the direct semantic guidance provided by semantic masking.

**Effect of Soft-Only Masking.** Soft masking, when used in isolation, encourages the model to learn from a wider range of masked positions. This approach does not specifically prioritize semantically significant

nodes or tokens. While this broader exposure can contribute to robust generalization capabilities, the absence of semantic information may limit the model's capacity to concentrate on key relationships. These key relationships are often critical for effectively tackling many graph-based tasks.

Overall we can see that both semantic and soft masking are required for any of the hyperparameters to achieve better performance.

## B  Node Importance and Semantic Masking Rationale

In our framework, semantic masking is guided by the concept of *node importance*, with **degree centrality** serving as a straightforward and effective measure. Degree centrality quantifies the number of edges incident to a node, providing a local measure of node connectivity. Formally, for a graph $G = (V, E)$, the degree centrality $C_D(v)$ of a node $v \in V$ is defined as:

$$C_D(v) = \deg(v) = |\{u \in V : (v, u) \in E\}|.$$

This measure reflects the immediate neighborhood of $v$, indicating its potential influence in local information propagation.

### B.1  Justification for Degree Centrality

Degree centrality is a strong proxy for node influence because highly connected nodes are generally more influential in propagating information and maintaining the overall connectivity of the graph. In many real-world datasets, these nodes are critical for ensuring that the graph remains cohesive, as their removal or perturbation can significantly disrupt network structure.

Masking nodes with high degree centrality compels the model to learn more robust representations. By targeting these *critical nodes*, we force the network to rely less on a few dominant nodes and instead capture diverse pathways and latent structural patterns. This idea aligns with notions of *network robustness* in graph theory, where the resilience of a network is evaluated by its ability to withstand the removal of key nodes.

### B.2  Comparison with Other Centrality Measures

While other centrality measures such as betweenness, closeness, and eigenvector centrality offer valuable insights into different aspects of node importance, degree centrality was chosen for its simplicity and computational efficiency.

### B.3  Betweenness Centrality

Betweenness centrality quantifies the frequency of a node appearing on shortest paths between other nodes. It is defined as:

$$C_B(v) = \sum_{s \neq v \neq t} \frac{\sigma_{st}(v)}{\sigma_{st}},$$

where $\sigma_{st}$ is the total number of shortest paths from node $s$ to node $t$, and $\sigma_{st}(v)$ is the number of those paths that pass through $v$. While informative, the computation of betweenness centrality typically scales as $\mathcal{O}(|V||E|)$, making it prohibitive for large graphs.

### B.4  Closeness Centrality

Closeness centrality measures the average distance from a node to all other nodes in the graph. It is defined as:

$$C_C(v) = \frac{1}{\sum_{u \in V} d(v, u)},$$

where $d(v, u)$ is the shortest-path distance between nodes $v$ and $u$. This measure reflects how quickly a node can access all other nodes in the network. However, its computation requires knowledge of all-pairs shortest paths, which can be computationally intensive for large networks.

### B.5 Eigenvector Centrality

Eigenvector centrality assigns relative scores to all nodes in the network based on the concept that connections to high-scoring nodes contribute more to the score of the node in question than equal connections to low-scoring nodes. It is defined as the principal eigenvector of the adjacency matrix of the graph. While theoretically appealing, the computation of eigenvector centrality involves solving an eigenvalue problem, which can be computationally expensive for large graphs.

Despite the theoretical advantages of alternative centrality measures, degree centrality offers a practical and computationally efficient approach for semantic masking in large-scale graphs. Its simplicity allows for real-time processing, making it an optimal choice for our semantic masking strategy.

## C   Justification for Combining GNNs and LMs

Let:

- $\mathcal{G} = (V, E)$ be a graph with a set of nodes $V$ and a set of edges $E$.

- $\mathbf{X} \in \mathbb{R}^{|V| \times d}$ be the node feature matrix, where $\mathbf{x}_i \in \mathbb{R}^d$ is the feature vector for node $v_i \in V$. Assume $\mathbf{x}_i$ contains or represents rich textual/semantic information.

- $\mathbf{A}$ be the adjacency matrix representing the edge set $E$.

- $\mathbf{h}_i \in \mathbb{R}^{d'}$ be the desired robust representation for node $v_i$.

### C.1   Graph Neural Network (GNN) Formulation (Structure-centric)

A typical GNN computes node representations iteratively. The representation of node $v_i$ at layer $l$, denoted $\mathbf{h}_i^{(l)}$, is a function of its previous representation $\mathbf{h}_i^{(l-1)}$ and the aggregated representations of its neighbors $\mathcal{N}(i)$:

$$\mathbf{h}_i^{(l)} = \text{UPDATE}^{(l)} \left( \mathbf{h}_i^{(l-1)}, \text{AGGREGATE}^{(l)} \left( \{ \mathbf{h}_j^{(l-1)} \mid j \in \mathcal{N}(i) \} \right) \right)$$

- **Focus:** The core operation AGGREGATE explicitly uses the graph structure (via $\mathcal{N}(i)$, derived from $\mathbf{A}$) to combine information from *neighbors*. Even with attention mechanisms like in Graph Attention Networks (GAT), where aggregation is weighted (e.g., $\sum_{j \in \mathcal{N}(i)} \alpha_{ij} \mathbf{W} \mathbf{h}_j^{(l-1)}$), the process is fundamentally driven by the *connectivity* defined in $E$.

- **Theoretical Limitation:** While GNNs capture structural patterns and local feature context effectively, the formulation primarily emphasizes how information propagates *along edges*. It may not inherently possess the mechanisms to deeply model complex, long-range semantic relationships *within* the feature vectors $\mathbf{x}_i$ themselves, especially if these relationships are not directly mirrored by the immediate graph topology. The paper notes GNNs primarily focus on structural information and local features, potentially overlooking rich semantic relationships.

### C.2   Language Model (LM) Formulation (Semantic-centric)

A pre-trained Language Model (LM) (like BERT) operates on sequences (text). When applied to node features $\mathbf{x}_i$ (assuming they are text or can be sequentialized), it produces a contextualized representation $\mathbf{b}_i$:

$$\mathbf{b}_i = \text{LM}(\text{sequence}(\mathbf{x}_i))$$

- **Focus:** LMs use mechanisms like self-attention over the input sequence. This allows them to model dependencies and contextual relationships between elements *within* the sequence$(\mathbf{x}_i)$, capturing nuanced semantic meaning. The paper highlights the LM's ability to capture contextual information.

- **Theoretical Limitation:** When applied node-by-node, the LM formulation inherently processes each $\mathbf{x}_i$ largely independently of the graph structure $E$. It lacks a direct mechanism to incorporate information about how node $v_i$ is connected to other nodes $v_j$ in the graph $\mathcal{G}$.

## C.3   Justification for Combination

The goal is to learn a node representation $\mathbf{h}_i$ that reflects *both* the structural role of $v_i$ in $\mathcal{G}$ (informed by node features $\mathbf{X}$) and is robust to the masking strategy employed during training. Our dual-branch architecture addresses this by combining complementary information sources:

- **GNN Branch (Structure/Feature-centric):** A GNN-based representation, $\mathbf{h}_i^{\text{GNN}} = f_{\text{GNN}}(\mathbf{A}, \mathbf{X})$, effectively encodes structural context and local feature information derived from the node features $\mathbf{X}$ and the adjacency matrix $\mathbf{A}$. However, relying solely on the GNN might be sensitive to the information disruption caused by masking nodes.

- **LM Branch (Masking State-centric):** Let $m_i$ represent the masking state of node $v_i$ (i.e., whether it is masked or not). Our LM branch takes a static token representation corresponding to $m_i$ (specifically, the strings "[MASK]" or "node feature" in our implementation) as input. **Crucially, it does *not* process the node features $\mathbf{X}_i$.** It produces a representation $\mathbf{b}_i = f_{\text{LM}}(\text{token}(m_i))$. The value of using a PLM here arises from: [label=()]

- leveraging the PLM's rich, pre-trained semantic space for initialization, and

- fine-tuning $\mathbf{b}_i$ during training to optimally represent the masking state $m_i$ specifically for effective fusion with the GNN branch via downstream task gradients.   This branch inherently ignores the graph structure $\mathbf{A}$ and the node features $\mathbf{X}$.

**Necessity for Combination:** Effectively learning from masked graph data requires understanding both the underlying graph structure/features and the implications of the masking itself. Neither branch alone captures this completely:

- The GNN branch leverages the graph context ($\mathbf{A}$, $\mathbf{X}$) but lacks a dedicated mechanism to represent the masking state $m_i$ itself in a nuanced way.

- Our LM branch provides a powerful, fine-tuned representation of the masking state $m_i$, leveraging pre-trained knowledge and architectural capacity, but ignores the actual graph context ($\mathbf{A}$, $\mathbf{X}$).

Therefore, a combined model is necessary to integrate these complementary perspectives.  The proposed dual-branch architecture formalizes this:

- One branch computes a structure-aware representation using the GNN (e.g., GAT): $\mathbf{h}_i^{\text{GNN}} = f_{\text{GNN}}(\mathbf{A}, \mathbf{X})$

- Another branch computes a masking-state-aware representation using the fine-tuned LM: $\mathbf{b}_i = f_{\text{LM}}(\text{token}(m_i))$

- These are fused via a dedicated network: $\mathbf{h}_i^{\text{final}} = \text{FUSE}(\mathbf{h}_i^{\text{GNN}}, \mathbf{b}_i)$

The combined representation $\mathbf{h}_i^{\text{final}}$ integrates the GNN's view of the graph structure $\mathbf{A}$ and node features $\mathbf{X}$ with the LM's optimized representation of the masking state $m_i$. This allows the model to potentially learn more robustly under the masking strategy by explicitly conditioning on both the graph context and a nuanced representation of whether a node's information is present or masked, aiming to overcome the limitations of each individual approach.

| Dataset | Nodes | Edges | Features | Classes |
|---|---|---|---|---|
| Cora | 2,708 | 10,556 | 1,433 | 7 |
| Citeseer | 3,327 | 9,104 | 3,703 | 6 |
| Pubmed | 19,717 | 88,648 | 500 | 3 |
| Amazon Photos | 7,650 | 238,612 | 745 | 8 |
| Amazon Computers | 13,752 | 491,722 | 767 | 10 |
| Coauthor CS | 18,333 | 163,788 | 6,805 | 15 |

Table 5: Dataset Details for the datasets used in the node classification tasks.

## D  Dataset Statistics

### D.1  Graph Tasks

Table 5 summarizes the key statistics for the benchmark datasets used in our node classification experiments. These datasets represent standard citation, co-purchase, and co-authorship networks commonly employed for evaluating Graph Neural Networks.

| Dataset | #Graphs | Avg. Nodes | Avg. Edges | Classes |
|---|---|---|---|---|
| COLLAB | 5,000 | 74.49 | 2457.78 | 3 |
| IMDB-B | 1,000 | 19.77 | 96.53 | 2 |
| PROTEINS | 1,113 | 39.06 | 72.82 | 2 |
| MUTAG | 188 | 17.93 | 39.58 | 2 |

Table 6: Dataset details for the datasets used in the Graph Classification tasks.

Table 6 summarizes the key statistics of the graph classification datasets used in our experiments. The datasets vary significantly in terms of the number of graphs, the average number of nodes and edges per graph, and the number of target classes. COLLAB consists of larger collaboration networks, while IMDB-B contains smaller social graphs. PROTEINS and MUTAG are bioinformatics datasets representing molecular structures and protein interactions. This diversity ensures a comprehensive evaluation across domains with different structural properties.

### D.2  GLUE Tasks

We perform grid search over three key variables: `pre_train_epochs`, `temperature`, and `finetune_epochs`. The specific values considered for each parameter are as follows:

- `temperature_values` = {0.3, 0.5, 0.7}

- `pretrain_epoch_values` = {5, 10}

- `finetune_epoch_values` = {5, 10}

Table 7 describes the hyperparameters selected via grid search for evaluating GMLM on GLUE.

Table 8 outlines the specific tasks from the GLUE benchmark selected for evaluating the language understanding capabilities of our model. It details the objective, input format, evaluation metric, and approximate size for each task.

| Task | Temperature | Pretrain Epochs | Finetune Epochs |
|------|-------------|-----------------|-----------------|
| WNLI | 0.3 | 10 | 5 |
| CoLA | 0.3 | 5 | 10 |
| STS-B | 0.7 | 5 | 10 |
| MRPC | 0.5 | 5 | 10 |
| SST-2 | 0.7 | 10 | 10 |
| RTE | 0.5 | 10 | 5 |

Table 7: Best hyperparameter settings for each GLUE task, determined via grid search.

| Task | Description | Input | Metric(s) | Size (Train/Dev) |
|------|-------------|-------|-----------|------------------|
| CoLA | **Corpus of Linguistic Acceptability**: Determine if a sentence is grammatically acceptable. | Single Sentence | Matthews Corr. | 8.5k / 1k |
| MRPC | **Microsoft Research Paraphrase Corpus**: Determine if two sentences are paraphrases of each other. | Sentence Pair | Accuracy / F1 | 3.7k / 0.4k |
| RTE | **Recognizing Textual Entailment**: Determine if a premise sentence entails a hypothesis sentence. | Sentence Pair | Accuracy | 2.5k / 0.3k |
| SST-2 | **Stanford Sentiment Treebank**: Determine the sentiment (positive/negative) of a single sentence (movie review). | Single Sentence | Accuracy | 67k / 0.9k |
| STS-B | **Semantic Textual Similarity Benchmark**: Predict a similarity score (1-5) between two sentences. | Sentence Pair | Pearson/Spearman Corr. | 5.7k / 1.5k |
| WNLI | **Winograd NLI**: A small textual entailment dataset focused on pronoun resolution challenges. | Sentence Pair | Accuracy | 0.6k / 0.1k |

Table 8: GLUE Benchmark Task Details. Train/Dev sizes are approximate.

