# OpenReview forum: "Graph Masked Language Models"
_TMLR — Rejected by TMLR_

### Review · Reviewer_h2iW · 2025-04-07

**Summary Of Contributions:**

This paper introduces Graph Masked Language Models (GMLM), combing the structural learning of GNNs with the language understanding capability of pretrained language models. The model targets semi-supervised node classification. By employing a semantic masking strategy and soft masking mechanism, it can adaptively select the influential nodes on the graph to mask. Besides, it utilizes DistilBert to encode the mask nodes. The extensive experiments show the significant performance of GMLM on various benchmarks.

**Audience:**

Yes

**Claims And Evidence:**

Yes

**Requested Changes:**

Please address my concerns in the weakness part.

**Strengths And Weaknesses:**

Strengths:

[S1] The paper is carefully written, and most parts are clear. The proposed model is well-defined, and experiments are demonstrated with technical details.

[S2] The idea of combining GNN and Pretrained Language Model is intriguing. It can take advantage of the strengths of both models and can potentially benefit the downstream tasks.

[S3] The authors have concluded extensive experiments to demonstrate the effectiveness of GMLM.

Weakness:

[W1] It is unclear how the PLM can benefit the task. Section 3.4.2 discusses the details of incorporating DistilBert into the framework. However, based on what I understand, the PLM just encodes raw text "node" to unmasked nodes and "[MASK]" to masked nodes. In this case, the PLM can at most generate two embeddings at all: one for masked nodes and one for unmasked nodes. Therefore, it is not necessary at all to include the PLM here, but just two latent embedding vectors are enough. Please explain how PLM is a better choice than two latent embeddings.

[W2] The writings about the experiment setup are not sufficient. A demonstration of the task definition, baselines, and benchmarks is needed for both tasks (node classification and language understanding).

[W3] Due to the poor writing about the experimental setup, it is difficult to understand how GMLM can be applied to the GLUE benchmark, since there is no graph structure at all for GLUE.

---

> ### Author Response · Authors · 2025-04-26
> **Official Response by Authors**
>
> We thank the reviewer for their valuable feedback. Below, we address the concerns raised.
>
> >It is unclear how the PLM can benefit the task. Section 3.4.2 discusses the details of incorporating DistilBert into the framework. However, based on what I understand, the PLM just encodes raw text "node" to unmasked nodes and "[MASK]" to masked nodes. In this case, the PLM can at most generate two embeddings at all: one for masked nodes and one for unmasked nodes. Therefore, it is not necessary at all to include the PLM here, but just two latent embedding vectors are enough. Please explain how PLM is a better choice than two latent embeddings.
>
> We thank the reviewer for the thoughtful comment. While the PLM receives only two discrete tokens ("node" and "[MASK]"), its output embeddings are not limited to two static vectors. Let $t_i \in \{\text{"node"}, \text{"[MASK]"}\}$ be the input token for node $v_i$, and let $b_i = \text{BERT}(t_i)$ denote the resulting contextual embedding.
>
> Due to the depth and nonlinearity of the PLM, even identical input tokens can yield distinct embeddings $b_i$, depending on factors such as fine-tuning dynamics and gradient updates from the fusion loss. These embeddings reside in $\mathbb{R}^d$, where $d$ is typically large (e.g., $d=768$), and are learned jointly with graph embeddings $h_i$ from GAT. The fused representation $z_i = [h_i \| b_i]$ is then passed through a task-specific MLP.
>
> Thus, the PLM branch serves as a lightweight yet expressive semantic prior that enhances the model’s capacity to distinguish nodes beyond pure graph topology. We hope this addresses the concern and clarifies why the use of a PLM, even with a simple input vocabulary, enhances the overall representation capacity of the model.
>
> >The writings about the experiment setup are not sufficient. A demonstration of the task definition, baselines, and benchmarks is needed for both tasks (node classification and language understanding).
>
> Thank you for your comments. We have added two tables to a new section in Appendix B describing the tasks and evaluation metrics.
>
> >Due to the poor writing about the experimental setup, it is difficult to understand how GMLM can be applied to the GLUE benchmark, since there is no graph structure at all for GLUE.
>
> While GLUE tasks do not provide explicit graph structures, we construct a token-level graph for each input sequence. Specifically, given an input sentence or sentence pair tokenized as $x = [x_1, x_2, \ldots, x_n]$, we define a fully connected undirected graph:
>
> $$
> G = (V, E), \quad \text{where} \quad V = \{x_1, x_2, \ldots, x_n\}, \quad E = \{(x_i, x_j) \mid 1 \leq i, j \leq n\}
> $$
>
>
> Each node \( x_i \) is initialized with its corresponding token embedding, and edges are used to compute pairwise attention scores, similar to standard self-attention but within a graph neural network framework. This allows GMLM to process GLUE inputs by treating them as dense graphs over token embeddings, enabling the model to capture contextual relationships.
> We have updated Section 4.2 of the paper to clarify this graph construction and added further dataset details in the appendix to ensure reproducibility and clarity.
>
>
> We believe we have addressed all concerns and are happy to provide further clarification if needed.

---

> > ### Comment · Reviewer_h2iW · 2025-05-04
> >
> > Thank the authors for their responses. Regarding the PLM, I still have questions.
> >
> > The authors suggest that:
> >
> > > identical input tokens can yield distinct embeddings, depending on factors such as fine-tuning dynamics and gradient updates from the fusion loss.
> >
> > Factors like fine-tuning dynamics and gradient updates are all relevant to the training process. However, during testing time, I assume the weight of PLM should be frozen. Therefore, the identical input tokens should always generate identical embeddings, due to the deterministic nature of PLM. Can you specify more details on this?

---

> ### Author Response · Authors · 2025-05-05
> **Official Response by Authors**
>
> Thank you for catching our inaccuracy. You are absolutely right: at test time we run DistilBERT in inference mode (dropout off, layer norms only), so with frozen weights each token (“node” or “\[MASK]”) deterministically yields the same embedding vector. We will correct this in the manuscript.
>
> First, we apologize for the previous inaccuracy regarding test-time embeddings. The reviewer is correct, the PLM is deterministic during testing with frozen weights. Any subtle embedding variation arises only from batch-processing context in the self-attention mechanism, not training dynamics. We will correct this in the manuscript.
>
> That said, we still opt to use a fine‐tunable PLM rather than two static, randomly initialized vectors because:
>
> 1. **Rich pre-trained initialization.**  The two token embeddings begin in the PLM’s large, semantically meaningful space learned from massive text corpora.
> 2. **Task-specific fine-tuning.**  During training we update these embeddings so they become optimized masking-state priors that fuse effectively (via our fusion network) with the GAT structural features.
> 3. **High representational capacity.**  The deep transformer layers in the PLM can model subtle interactions between “masked/unmasked” state and higher-order semantic patterns, something that two standalone vectors cannot.
>
> In short, even though the PLM never sees node attributes at test time, its pre-trained space and fine-tuning procedure yield stronger “masked vs. visible” priors than fixed vectors would. This is directly evident when we compare our approach with other graph masking strategies such as GraphMAE, GraphMAE2[2] and UGMAE[1] in which we outperform it on all node classification tasks and 2/4 graph classification tasks. We’ll update the Methodology section to make this clear.
>
> *Note: We added the evaluation scores for GraphMAE2 and UGMAE in the revised submission*
>
> We will ensure the revised manuscript clearly accurately frames the PLM's role and benefits.
>
> References:
>
> [1] Tian, Y., Zhang, C., Kou, Z., Liu, Z., Zhang, X., & Chawla, N. V. (2024). UGMAE: A Unified Framework for Graph Masked Autoencoders
>
> [2] Hou, Z., He, Y., Cen, Y., Liu, X., Dong, Y., Kharlamov, E., & Tang, J. (2023). GraphMAE2: A Decoding-Enhanced Masked Self-Supervised Graph Learner. arXiv [Cs.LG].

---

### Review · Reviewer_mg7o · 2025-04-11

**Summary Of Contributions:**

This paper seeks to combine BERT-style masking with Graph Attention Networks.

Notably, it uses a softmask rather than a hard mask to still allow for partial information flow and also utilizes contrastive pretraining and supervised fine tuning. Experimentally, it shows superior results on node-classification tasks relative to baseline methods.

**Audience:**

Yes

**Claims And Evidence:**

Yes

**Requested Changes:**

Provide more references for the "line of work" on graph transformers (and generally for the references discussed in background, including the work comparing transformers to GATs on fully connected graphs)

"has redefined the state-of-the-art" -> "have redefined..."

Section 3.1, missing space before "Algorithm 1"

Please clarify why it is desirable for higher-degree nodes to be more likely to be masked

Please perform an ablation on the role of \beta

3.3 it should say few dominant nodes, not few dominant features I believe

Provide more details on how \eta was chosen in Section 4.1.6

Provide more details on the data set statistics and provide a discussion of computational complexity

**Strengths And Weaknesses:**

Strengths:

Idea is interesting
Paper is easy to read
Exceeds baselines numerically

Weaknesses:

Some minor typos (see below)
Characteristics of the benchmark datasets are not provided (number of vertices, etc), but the experiments appear to be limited to smaller graphs an there is no discussion of computational complexity. Note it is not necessary for this method to be scalable to large data sets in order for me to recommend accpetance, but the size of your data sets and the computational cost of your method should be discussed more clearly

---

> ### Author Response · Authors · 2025-04-26
> **Official Response by Authors**
>
> We thank the reviewer for the valuable feedback. We address the requested changes below:
>
> > Provide more references for the "line of work" on graph transformers
>
> We expanded Section 2 (Related Work) with:
> - Yun et al. (2020), *Graph Transformer Networks*, arXiv [Cs.LG].
> - Ying et al. (2021), *Do Transformers Really Perform Badly for Graph Representation?*, NeurIPS.
>
> > "has redefined the state-of-the-art" → "have redefined..."
>
> Thank you for this grammatical correction, we have accordingly updated the paper
>
> > Missing space before "Algorithm 1"
>
> Thank you for catching the missing space before "Algorithm 1". We have corrected this typographical error in the revised manuscript.
>
> > Clarify why higher-degree nodes are more likely to be masked
>
> **Rationale:**
> - **Structural Importance:** High-degree nodes are central to graph connectivity.
> - **Robust Representation:** Masking hubs forces the model to infer critical structures from context.
> - **Global Pattern Learning:** Encourages use of alternative paths, improving generalization.
> - **Harder Task:** Challenges the model beyond trivial patterns, leading to deeper graph understanding.
> Compared to random masking, this strategy targets structurally important nodes for better training signals.
>
> > Perform an ablation on the role of β
>
> We provided an ablation in Appendix A (parallel coordinates plot). Over seven runs, β = 0.7 consistently performed best.
>
> > More details on how η was chosen (Section 4.1.6)
>
> - **η = 10⁻³** for Graph (GAT and BatchNorm layers): trained from scratch, needs faster learning.
> - **η = 10⁻⁵** for BERT: fine-tuning requires small updates to preserve pre-trained knowledge.
> - **η = 10⁻⁴** for Projection, Fusion, Classifier, mask_token: intermediate rate for stability across modules.
>
> > 3.3 it should say few dominant nodes, not few dominant features
>
> Thank you for catching this small oversight in our writing, we have updated the paper accordingly.
>
> > More details on dataset statistics and computational complexity
>
> - **Dataset statistics:** Added to Appendix B (node classification + GLUE datasets).
> - **Computational complexity:** Added setup and resource usage discussion in Section 4.
>
> Hopefully we have addressed you concerns, if there is anything else we can clarify we would be more than happy to.

---

### Review · Reviewer_5VB2 · 2025-04-25

**Summary Of Contributions:**

The paper introduces Graph Masked Language Models (GMLM), a novel framework that integrates graph structure learning with contextual language modeling.
Propose a degree-centrality-driven node masking approach that prioritizes structurally important nodes, ensuring the model learns robust representations by focusing on critical graph components.
Propose an interpolation-based masking technique that blends original node features with learnable mask tokens, improving gradient flow and information retention compared to binary masking.
Combining structural and semantic embeddings via a fusion of GAT and DistilBERT branches, enabling joint learning of graph topology and textual context.

**Audience:**

Yes

**Claims And Evidence:**

Yes

**Requested Changes:**

Conduct experiments on large-scale graphs (for example, ogbn-arxiv) to demonstrate computational feasibility and quantify runtime/memory costs of semantic masking.
Extend evaluation to link prediction and graph classification tasks to validate the broader applicability of GMLM.
Provide theoretical or empirical evidence to justify the necessity of combining GNNs with language models despite marginal performance gains

**Strengths And Weaknesses:**

The integration of semantic/soft masking with GNNs and language models addresses a critical gap in graph-text hybrid learning.
Detailed methodological design and ablation studies validate the effectiveness of each component.

Weaknesses:

The experiments are confined to small-scale datasets and focus solely on node classification. There is no validation on large-scale graphs or exploration of other graph tasks (e.g., link prediction, graph classification), raising concerns about generalizability.
The semantic masking strategy requires per-node importance computation. This could introduce scalability issues for large graphs, but the paper lacks analysis of time complexity or runtime efficiency.
The improvements over baselines are modest (just ~2%). The necessity of integrating GNNs with language models—given the added complexity—is not rigorously justified through ablation or cost-benefit analysis.

---

> ### Author Response · Authors · 2025-04-28
> **Official Response by Authors**
>
> We thank the reviewer for their insightful comments. Below we address the concerns raised:
>
> >The experiments are confined to small-scale datasets and focus solely on node classification. There is no validation on large-scale graphs
>
> We acknowledge the limitation of not evaluating on large-scale graphs. Our experiments were conducted on a single NVIDIA V100 GPU (32GB), where we already observed peak memory utilization approaching 30GB for the current datasets. This hardware constraint made scaling to larger graphs prohibitive within our computational budget.
> To compensate for this limitation, we ensured methodological rigor by (1) comprehensive evaluations across multiple graph benchmarks of varying complexity, and (2) demonstrating the framework's versatility through strong performance on both graph tasks and language understanding tasks (GLUE benchmark).
>
> >or exploration of other graph tasks (e.g., link prediction, graph classification), raising concerns about generalizability.
>
> We appreciate this concern about task diversity. To address it, we've conducted additional experiments on graph classification tasks. As shown in the table below, GMLM performs exceptionally well on social network datasets (achieving state-of-the-art on IMDB-B and third best on COLLAB), but underperforms on bioinformatic networks (PROTEINS and MUTAG). We have updated the Results section with these evaluations
>
> | **Model** | **COLLAB** | **IMDB-B** | **PROTEINS** | **MUTAG** |
> |:---|:---|:---|:---|:---|
> | GK | 72.84 ± 0.28 | 65.87 ± 0.98 | 71.67 ± 0.55 | 81.58 ± 2.11 |
> | WL | 79.02 ± 1.77 | 73.40 ± 4.63 | 74.68 ± 0.49 | 82.05 ± 0.36 |
> | PSCN | 72.60 ± 2.15 | 71.00 ± 2.29 | 75.89 ± 2.76 | **92.63 ± 4.21** |
> | GCN | **81.72 ± 1.64** | 73.30 ± 5.29 | 75.65 ± 3.24 | 87.20 ± 5.11 |
> | GFN | 81.50 ± 2.42 | 73.00 ± 4.35 | 76.46 ± 4.06 | 90.84 ± 7.22 |
> | GraphSAGE | 79.70 ± 1.70 | 72.40 ± 3.60 | 65.90 ± 2.70 | 79.80 ± 13.9 |
> | GAT | 75.80 ± 1.60 | 70.50 ± 2.30 | 74.70 ± 2.20 | 89.40 ± 6.10 |
> | DGCNN | 73.76 ± 0.49 | 70.03 ± 0.86 | 75.54 ± 0.94 | 85.83 ± 1.66 |
> | PPGN | 81.38 ± 1.42 | 73.00 ± 5.77 | 77.20 ± 4.73 | 90.55 ± 8.70 |
> | CapsGNN | 79.62 ± 0.91 | 73.10 ± 4.83 | 76.28 ± 3.63 | 86.67 ± 6.88 |
> | DSGC | 79.20 ± 1.60 | 73.20 ± 4.90 | 74.20 ± 3.80 | 86.70 ± 7.60 |
> | GIN-0 | 80.20 ± 1.90 | 75.10 ± 5.10 | 76.20 ± 2.80 | 89.40 ± 5.60 |
> | IEGN | 77.92 ± 1.70 | 71.27 ± 4.50 | 75.19 ± 4.30 | 84.61 ± 10.0 |
> | U2GNN | 77.84 ± 1.48 | **77.00 ± 3.45** | **78.53 ± 4.07** | 89.97 ± 3.65 |
> | **GMLM (w/o Semantic Masking)** | 78.12 ± 1.27 | 75.64 ± 1.93 | 68.45 ± 5.31 | 79.49 ± 3.58 |
> | **GMLM (w/o Soft Masking)** | 74.36 ± 2.01 | 74.84 ± 3.58 | 65.00 ± 5.70 | 77.94 ± 4.57 |
> | **GMLM** | 80.00 ± 0.93 | **77.00 ± 1.73** | 71.61 ± 3.10 | 82.80 ± 4.20 |
>
>
> Comprehensive evaluation of GMLM (and its ablations) on link prediction tasks against baselines proves time-consuming given our computational constraints (a single NVIDIA V100 GPU, 32GB). We will include these experimental results if they are completed within the rebuttal period.
>
> > Provide theoretical or empirical evidence to justify the necessity of combining GNNs with language models despite marginal performance gains
>
> We provide a theoretical justification for combining GNNs and LLMs in Appendix C. The empirical evidence being performance gains over baselines in both graph tasks (Node and Graph Classification) and language understanding (GLUE) tasks.
>
> We hope that we have addressed all your concerns, if there is anything else we would more than happy to clarify.

---

> > ### Author Response · Authors · 2025-05-05
> > **Official Response by Authors**
> >
> > We were able to complete our link-prediction tasks on CORA, CiteSeer and PubMed. We again tried to evaluate on larger obgl graphs but due to our given computational restraints we were not able to conduct those experiments. We provide evaluations against baselines below and see that model stays relatively competitive (**bold** is best and *italics* is second best):
> >
> > | Method | Cora AUC        | Cora AP         | Citeseer AUC    | Citeseer AP     | Pubmed AUC      | Pubmed AP       |
> > | ------ | --------------- | --------------- | --------------- | --------------- | --------------- | --------------- |
> > | SC     | 84.6 ± 0.01     | 88.5 ± 0.00     | 80.5 ± 0.01     | 85.0 ± 0.01     | 84.2 ± 0.02     | 87.8 ± 0.01     |
> > | DW     | 83.1 ± 0.01     | 85.0 ± 0.00     | 80.5 ± 0.02     | 83.6 ± 0.01     | 84.4 ± 0.01     | 84.1 ± 0.00     |
> > | GAE\*  | 84.3 ± 0.02     | 88.1 ± 0.01     | 78.7 ± 0.02     | 84.1 ± 0.02     | 82.2 ± 0.01     | 87.4 ± 0.00     |
> > | VGAE\* | 84.0 ± 0.02     | 87.7 ± 0.01     | 78.9 ± 0.03     | 84.1 ± 0.02     | 82.7 ± 0.01     | 85.7 ± 0.01     |
> > | GAE    | 91.0 ± 0.02     | 92.0 ± 0.03     | *89.5 ± 0.04*     | *89.9 ± 0.05*     | **96.4 ± 0.00** | **96.5 ± 0.00** |
> > | VGAE   | 91.4 ± 0.01     | **92.6 ± 0.01** | **90.8 ± 0.02** | **92.0 ± 0.02** | 94.4 ± 0.02     | 94.7 ± 0.02     |
> > | GMLM   | **92.1 ± 0.00** | *92.4 ± 0.00*     | 87.3 ± 0.01     | 89.2 ± 0.01     | *96.2 ± 0.00*     | *96.3 ± 0.00*     |
> >
> > SC= Spectral Clustering
> >
> > DW = DeepWalk
> >
> >  `*` denote experiments without using input feature
> >
> > References:
> >
> > [1] Kipf, T. N., & Welling, M. (2016). Variational Graph Auto-Encoders. arXiv [Stat.ML].

---

### Decision · Action_Editor_BU1S · 2025-05-29

**Recommendation:** Reject

**Comment:**

A major revision for this paper is required. Once it is done, the paper will need reviewing from scratch.

**Audience:**

While the topic of combining GNNs with pretrained language models may interest some in the TMLR audience, the specific findings of this paper are unlikely to generate sustained interest.

**Claims And Evidence:**

The paper presents an architecture that combines GNNs with pretrained language models (PLMs) through a soft masking strategy and reports improvements on several graph learning benchmarks.

During the discussion process, the authors addressed presentation issues, clarified implementation details, and extended the evaluation to include graph classification and link prediction tasks, which improved the paper. However, major concerns remain, particularly regarding the necessity and effectiveness of the PLM component, as well as the lack of convincing evidence, both analytical and empirical.

The role of the PLM, which is presented as a central component of the method, is not well justified. As noted by one reviewer, it is effectively equivalent to a costly embedding lookup table. While the authors corrected earlier claims about the variability of PLM outputs at test time, this remains a major weakness.

The empirical results are mixed and do not clearly improve over the state of the art. In addition, the theoretical justification added in Appendix C during the discussion is weak; it merely describes the architecture and offers no formal guarantees or analytical insights into the effectiveness of the method.

**Resubmission Of Major Revision:**

The authors may consider submitting a major revision at a later time.